# Vesicular Transport Mediated by Endoplasmic Reticulum Stress Sensor BBF2H7 Orchestrates Melanin Production During Melanogenesis

**DOI:** 10.3390/ijms27010501

**Published:** 2026-01-03

**Authors:** Giang Huy Phan, Kenshiro Fujise, Kazunori Imaizumi, Atsushi Saito

**Affiliations:** 1Department of Frontier Science and Interdisciplinary Research, Faculty of Medicine, Kanazawa University, 13-1 Takara-Machi, Kanazawa 920-8640, Ishikawa, Japan; 2The Osaka Medical Research Foundation for Intractable Diseases, 3-7-11, Minamisumiyoshi, Sumiyoshi-Ku, Osaka 558-0041, Osaka, Japan; 3Department of Child Development and Molecular Brain Science, United Graduate School of Child Development, Osaka University, Suita 565-0871, Osaka, Japan

**Keywords:** melanogenesis, tyrosinase, endoplasmic reticulum stress, BBF2H7, Sec23a, COPII vesicle, protein transport

## Abstract

The synthesis of the melanin pigment in melanocytes plays a crucial role in protecting the body from ultraviolet radiation. Tyrosinase, a key enzyme in melanogenesis, catalyzes the conversion of tyrosine to melanin in the melanosomes of melanocytes. During melanogenesis, Tyrosinase is abundantly synthesized in the lumen of the endoplasmic reticulum (ER) and subsequently transported from the ER to the melanosomes via the Golgi apparatus. In the present study, we demonstrate that Box B-binding factor 2 human homolog on chromosome 7 (BBF2H7), an ER-resident transmembrane transcription factor that functions as an ER stress sensor, is activated by mild ER stress caused by abundant Tyrosinase synthesis. Activated BBF2H7 enhances COPII-mediated anterograde transport by inducing the expression of Sec23a, which is a COPII component and transcriptional target of BBF2H7. Loss of BBF2H7 attenuates the transport of Tyrosinase, leading to its accumulation in the ER lumen and reduced melanin production. Restoration of BBF2H7 or Sec23a expression in *Bbf2h7*-deficient melanocytes rescues anterograde transport of Tyrosinase from the ER and melanin pigmentation. Collectively, these findings reveal that the BBF2H7-Sec23a axis is essential for the ER-to-melanosome transport of Tyrosinase and subsequent melanin synthesis. Thus, it may be a prospective therapeutic target for disorders related to melanin pigmentation.

## 1. Introduction

The skin is a composite tissue consisting of the epidermis, dermis, and hypodermis (fat, sebaceous glands, nails, sweat glands, and hair) [1]. As the interface between the external environment and the body, the skin has defense mechanisms that protect against continuous external stimuli. The outermost layer, or epidermis, is subdivided into four layers from the dermal side: the basal layer, spinous layer, granular layer, and stratum corneum [2,3]. These layers contain various cell types, including melanocytes, keratinocytes, and immune cells [4,5]. Among them, melanocytes play a crucial role in protecting the skin from ultraviolet (UV) radiation due to the synthesis and pigmentation of melanin [6]. While melanin protects against UV radiation, its excessive accumulation causes hyperpigmentation disorders such as solar lentigines, freckles, and melasma [7]. On the other hand, insufficient melanin pigmentation due to impaired melanogenesis leads to depigmentation disorders, including albinism and vitiligo [8,9]. Current treatments for these skin disorders are largely symptomatic and typically require long-term management, such as sun avoidance and sunscreen application (hyperpigmentation) or phototherapy combined with topical corticosteroids (hypopigmentation).

Melanogenesis is enhanced in melanocytes and stimulated by several factors, including α-melanocyte-stimulating hormone (α-MSH) [10]. Upon UV exposure, keratinocytes release α-MSH, which binds to the melanocortin 1 receptor on the plasma membrane of melanocytes. This binding induces the expression of the microphthalmia-associated transcription factor (MITF) [11,12]. MITF promotes the transcription of the *Tyrosinase* gene [13], leading to the exponential synthesis of the Tyrosinase protein in the endoplasmic reticulum (ER) lumen. The synthesized Tyrosinase is subsequently transported to melanosomes, which are specialized organelles responsible for melanin production [14,15]. Tyrosinase catalyzes the hydroxylation of tyrosine to DOPA and its subsequent conversion to dopaquinone in melanosomes [16]. Dopaquinone is then converted to melanin through autooxidation [17]. This series of steps indicates that the ER function ensures large-scale protein synthesis, and the efficient transport of Tyrosinase protein from the ER to melanosomes is essential for melanin production. Indeed, α-MSH stimulation causes ER stress, suggesting that the exponential synthesis of Tyrosinase increases the burden on the ER [18]. ER stress triggered by increased Tyrosinase synthesis activates the three major ER stress sensors: inositol-requiring enzyme 1 (IRE1) [19], activating transcription factor 6 (ATF6) [20], and protein kinase R-like ER kinase [21]. These sensors induce an unfolded protein response (UPR) to enhance the protein folding capacity in the ER and manage the massive amount of Tyrosinase synthesis present [18,22,23]. However, the regulatory mechanisms underlying protein transport from the ER to melanosomes during melanogenesis remain poorly understood.

Although the UPR has been historically considered a pathway dedicated to resolving ER stress [22,23], accumulating evidence indicates that it also regulates diverse biological processes, including cell differentiation [24,25,26,27,28,29], cell proliferation [30,31], lipid metabolism [32,33,34], glucose metabolism [35], and cellular senescence [36,37,38]. Box B-binding factor 2 human homolog on chromosome 7 (BBF2H7), also known as cyclic AMP response element-binding protein 3-like 2 (CREB3L2), is a type II transmembrane protein. Under normal conditions, BBF2H7 is localized at the ER membrane; the C-terminal domain targets the ER lumen; and the N-terminal domain targets the cytoplasm, which contains a basic leucine zipper DNA-binding domain [39]. Although the detailed mechanisms are still unknown, BBF2H7 senses ER stress and is activated as a transcription factor via regulated intramembrane proteolysis. Previous studies have shown that BBF2H7 not only senses pathological ER stress induced by the accumulation of unfolded or misfolded proteins, but is also activated as a transcription factor under physiological ER stress due to the increased burden on the ER from the exponential and massive synthesis of specific proteins [25]. Activated BBF2H7 directly induces the expression of *Sec23a*, which is a key component of COPII transport vesicles responsible for anterograde trafficking from the ER to the Golgi apparatus, thereby promoting export of proteins from the ER [25]. Another target gene of BBF2H7 is *Atf5* [40]. The upregulation of ATF5 by BBF2H7 suppresses apoptosis. From this, the downregulation of BBF2H7 leads to severe chondrodysplasia due to the accumulation of cartilage matrix proteins in the ER [25], suggesting that the regulation of protein transport by the BBF2H7-Sec23a pathway is important for maintaining biological functions. Given this unique function, it is plausible that the BBF2H7-Sec23a axis may regulate the export of abundant Tyrosinase synthesized in the ER during melanogenesis. However, the molecular mechanisms linking BBF2H7 activity to melanin production remain unknown.

In this study, we discovered that BBF2H7, activated in response to physiological ER stress during melanogenesis, facilitates the export of Tyrosinase from the ER by promoting its loading into COPII vesicles, and consequently accelerating melanin production. This discovery sheds light on the molecular mechanism underlying melanogenesis from the perspective of protein transport, which may lead to an understanding of the pathogenesis of melanin pigmentation disorders.

## 2. Results

### 2.1. BBF2H7 Is Activated During Melanogenesis

BBF2H7 is an ER-resident transmembrane transcription factor that contains a basic leucine zipper DNA-binding domain in its cytoplasmic region (Figure 1A). We previously demonstrated that BBF2H7 is cleaved by site-1 and site-2 proteases (S1P and S2P) in response to ER stress [39]. During ER stress, BBF2H7 is translocated to the Golgi apparatus and undergoes regulated intramembrane proteolysis mediated by S1P and S2P. The generated cytoplasmic N-terminal fragment acts as a transcription factor [39]. One of the transcriptional targets of BBF2H7 is *Sec23a*, which is a key component of COPII vesicles that promotes anterograde protein transport from the ER to the Golgi apparatus [25]. Although the direct association between BBF2H7 and pathogenesis is under investigation, the public gene expression dataset suggests that BBF2H7 is involved in melanin pigmentation disorders. Dr. Yvonne Mica and colleagues obtained RNAs from melanocytes derived from human embryonic stem cells and melanocytes derived from pluripotent stem cells in patients with oculocutaneous albinism (Hermansky–Pudlak syndrome). They performed a genome-wide analysis of gene expression to compare the expression levels of various genes in normal and patient-derived melanocytes (Gene Expression Omnibus accession number GSE45226) [41]. Based on this data, a relatively low expression level of *Bbf2h7* was identified in patient-derived melanocytes, compared to those derived from human embryonic stem cells (Figure 1B). The level of Tyrosinase expression was not reduced in the patients, and its levels were uncorrelated with those of BBF2H7 (Appendix A). Pathway and gene ontology analyses using this dataset revealed a strong association between oculocutaneous albinism and intracellular vesicle-mediated protein transport (Figure 1C,D and Appendix A). A low expression level of *Bbf2h7* has also been reported in lesional skin from patients with several types of vitiligo [42]. In addition to the abundant synthesis and proper folding of Tyrosinase in the ER lumen, the efficient transport of Tyrosinase from the ER to melanosomes is also a critical step in melanogenesis. Therefore, we examined whether BBF2H7-dependent protein transport contributes to melanin production.

ER stress is caused by the over-synthesis of Tyrosinase, and we previously demonstrated that this occurs due to the treatment of the B16 melanoma cell line with α-MSH [18]. Since BBF2H7 is activated in response to ER stress, we used this model to investigate the relationship between BBF2H7 and melanogenesis. Treatment with α-MSH increased the accumulation of melanin granules in B16 cells (Figure 1E–G). Consistent with previous findings, α-MSH induced the phosphorylated form (active form) of IRE1 in parallel with exponential Tyrosinase synthesis (Figure 1H) [18]. Following stimulation of α-MSH, Tyrosinase was localized to melanosome-like cytoplasmic puncta (Figure 1I). The production of the BBF2H7 N-terminal fragment (acts as a transcription factor) was also enhanced, followed by its accumulation in the nucleus (Figure 1H,I). We previously revealed that BBF2H7 directly binds to the cyclic AMP response element (CRE)-binding site within the *Sec23a* promoter, resulting in the induction of Sec23a expression in chondrocytes [25]. We also confirmed the direct binding of BBF2H7 to the *Sec23a* promoter in a B16 melanoma cell line treated with α-MSH (Appendix A). The induction of Sec23a was observed, coinciding with the increased level of the N-terminal fragment. Together, these results suggest that BBF2H7 acts as a transcription factor, promoting Sec23a expression in response to ER stress caused by the elevated synthesis of Tyrosinase during α-MSH-induced melanogenesis.

### 2.2. Activation of BBF2H7 Is Necessary for Vesicular Transport of Tyrosinase

Sec23a plays a central role in the assembly of COPII vesicles by recruiting other components, including the outer-coat protein Sec31a. Thus, vesicle formation is completed to enable transport of anterograde from the ER to the Golgi apparatus [43,44]. Since Sec23a expression was upregulated during melanogenesis, we examined COPII vesicle formation after α-MSH treatment. Under basal conditions, Sec31a was diffusely distributed throughout the cytoplasm of B16 cells (Figure 2A). Upon α-MSH treatment, Sec31a exhibited punctate localization, attributed to COPII vesicles and their recruitment to ER exit sites. Immunoreactivity of Tyrosinase partially overlapped with Sec31a-positive puncta in α-MSH-treated cells. Since Tyrosinase proteins rapidly move from the ER into melanosomes, most Tyrosinase-positive puncta in the cytoplasm are considered melanosomes. By contrast, it is thought that Sec31a- and Tyrosinase-positive puncta are COPII vesicles, containing the Tyrosinase protein. Thus, we concluded that Tyrosinase is exported from the ER via COPII-dependent trafficking. To determine whether the BBF2H7-Sec23a pathway is required for Tyrosinase transport, we knocked down *Bbf2h7* or *Sec23a* in B16 cells treated with α-MSH. The knockdown effectively reduced protein levels of BBF2H7 (si*Bbf2h7*) and Sec23a (si*Bbf2h7* or si*Sec23a*) without altering the total amount of Tyrosinase (Figure 2B). In these knockdown cells, immunoreactivity of Tyrosinase was diffusely localized in the ER, in contrast to the punctate distribution in cells transfected with non-targeting siRNA (Figure 2C,D).

### 2.3. Inhibition of BBF2H7-Mediated Vesicular Transport Suppresses Melanin Production

We then investigated the effects of *Bbf2h7* or *Sec23a* deficiency on melanin production. Phase-contrast imaging revealed a significant reduction in melanin granules in *Bbf2h7*- or *Sec23a*-knockdown cells during melanogenesis (Figure 3A). Consistently, the color of the cell pellets was noticeably lighter in these knockdown cells than in those transfected with non-targeting siRNA (Figure 3B). Quantification of melanin extracted from each sample showed a decrease in melanin content upon the suppression of *Bbf2h7* or *Sec23a* expression (Figure 3C). Together with the data in Figure 2, these findings indicate that downregulation of the BBF2H7-Sec23a pathway disrupts the proper export of Tyrosinase from the ER, eventually attenuating melanin production.

We revealed that the downregulation of the ER folding capacity is caused by inhibition of the IRE1 or ATF6 pathway, which suppresses melanin production [18]. Thus, we checked the crosstalk between the BBF2H7 pathway and the IRE1 or ATF6 pathway. The expression levels of IRE1 and ATF6 were not significantly changed by the knockdown of *Bbf2h7* (Appendix A), suggesting that there is no crosstalk between these pathways. Additionally, we carried out a melanin content assay using *Bbf2h7*-knockdown B16 cells treated with 4μ8c (a specific inhibitor of IRE1) or siRNA targeting *Atf6*. We confirmed the reduction in the spliced form of X-box binding protein 1, the downstream molecule IRE1, due to treatment with 4μ8c (Appendix A). The expression level of ATF6 was suppressed by the knockdown of *Atf6* (Appendix A). Melanin production was not synergistically reduced by the suppression of both the BBF2H7 pathway and the IRE1 or ATF6 pathway (Appendix A). These data indicate that the inhibition of either anterograde transport of Tyrosinase (BBF2H7 pathway) or protein quality control in the ER lumen (IRE1 and ATF6 pathways) is sufficient to attenuate melanin production.

### 2.4. Restoration of the BBF2H7-Sec23a Pathway Promotes Vesicular Transport of Tyrosinase and Melanin Production

To further examine the roles of BBF2H7 and Sec23a in melanogenesis, we tried to utilize *Bbf2h7*-deficient (*Bbf2h7*^−/−^) mice. The mice exhibited severe chondrodysplasia caused by the accumulation of cartilage matrix proteins in the ER lumen due to the downregulation of protein trafficking from the ER [25]. Although *Bbf2h7*^−/−^ mice are born, they die shortly after birth due to respiratory failure associated with an immature chest cavity. Therefore, we performed histological analyses using embryonic day 18.5 wild-type and *Bbf2h7*^−/−^ mice. No significant differences were observed in the overall layer structure or cellular composition of the abdominal epidermis and dermis between wild-type and *Bbf2h7*^−/−^ mice (Figure 4A). Melanin pigments detected by Fontana–Masson staining were observed scarcely in the abdominal skin of both wild-type and *Bbf2h7*^−/−^ mice, likely due to a lack of external stimuli such as UV exposure, which is generally required to assess melanin pigments in murine skin [45,46,47]. Thus, we extracted primary melanocytes from wild-type and *Bbf2h7*^−/−^ mice to evaluate melanin production in vitro. In wild-type melanocytes, α-MSH stimulation increased the levels of N-terminal fragments of BBF2H7 and Sec23a, but not *Bbf2h7*^−/−^ melanocytes (Figure 4B). Expression of Tyrosinase induced by α-MSH was unaffected by *Bbf2h7* deficiency. In wild-type melanocytes, α-MSH promoted the punctate localization of Sec31a and Tyrosinase, with partial colocalization (Figure 4C). In contrast, *Bbf2h7*^−/−^ melanocytes exhibited diffuse mislocalization of both proteins following α-MSH treatment. Furthermore, the increase in melanin production and melanin granule formation by α-MSH observed in wild-type cells was reversed in *Bbf2h7*^−/−^ melanocytes (Figure 4D,E). These data suggest that the anterograde transport of Tyrosinase via COPII vesicles is impaired in *Bbf2h7*^−/−^ melanocytes during melanogenesis, resulting in diminished melanin accumulation.

To restore Tyrosinase transport from the ER, we overexpressed BBF2H7 or Sec23a in *Bbf2h7*^−/−^ melanocytes. We observed an increase in the amount of Sec23a due to its exogenous expression. The induction of Sec23a was also observed in *Bbf2h7*^−/−^ melanocytes; they exogenously expressed BBF2H7 without affecting Tyrosinase protein levels (Figure 5A). Overexpression of these proteins rescued the mislocalization of Sec31a and Tyrosinase, with partial colocalization (Figure 5B). Restoration of Sec23a expression and COPII-mediated anterograde transport also enhanced melanin production and melanin granule formation in *Bbf2h7*^−/−^ melanocytes treated with α-MSH (Figure 5C,D). Taken together, we concluded that BBF2H7-mediated Sec23a induction and subsequent COPII vesicle formation are essential for the ER export of Tyrosinase and proper melanin production.

## 3. Discussion

We previously demonstrated that protein quality control in the ER, which ensures the large-scale synthesis of Tyrosinase, is crucial for melanogenesis [18]. In this study, we also reveal that the anterograde transport of Tyrosinase from the ER plays an indispensable role in melanin production. Anterograde transport is promoted by the activation of BBF2H7 in response to mild ER stress due to the massive synthesis of Tyrosinase. Our findings provide new insights into the molecular mechanisms underlying melanogenesis from the perspective of ER functions responsible for protein synthesis and trafficking. A key concept is that ER stress sensors orchestrate different biological processes. The downregulation of ER stress sensors such as IRE1 and ATF6 impairs ER protein quality control, resulting in a decreased Tyrosinase level and a reduction in melanin production [18]. On the other hand, it has been reported that the expression of genes involved in protein secretion pathways and membrane lipid biosynthesis is controlled by several types of ER stress sensors, including BBF2H7, IRE1, and ATF6 [32,48,49]. As the regulation of protein secretion machinery and membrane biogenesis is significant for COPII vesicle budding and formation, understanding the coordinated crosstalk between these UPR-related molecules is critical for understanding the mechanisms driving melanogenesis.

The change in melanin content in this study was due to the regulation of the protein transport system, which may lead to therapeutic approaches for skin disorders related to melanin pigmentation. For example, the suppression of vesicular transport to prevent the anterograde transport of Tyrosinase could be applicable to the treatment of hyperpigmentation. Additionally, the promotion of Tyrosinase export from the ER also has potential as a therapeutic target for melanin depigmentation. Several types of genetically mutated proteins can exert their functions when properly delivered to their target regions [50,51,52]. Although the strategy for targeting protein transport may be unsuitable for disorders characterized by the complete loss of Tyrosinase activity, such as oculocutaneous albinism type 1A (OCA1A), it might be effective for patients with OCA1B, who have partially retained Tyrosinase activity [9]. This can be achieved by accelerating the transport of Tyrosinase from the ER to the melanosomes. Analyses using cells and tissues derived from patients with these disorders can help understanding the relationship between BBF2H7-mediated protein transport and melanin production, which may further clarify the therapeutic potential of modulating vesicular transport in pigmentation disorders.

BBF2H7 is activated as a transcription factor via the cleavage by S1P and S2P (Figure 1A). Several compounds have been shown to potentially alter the activity of BBF2H7. For example, the previous report indicated that Nelfinavir inhibits S2P activity [53]. However, treatment with Nelfinavir may have unintended consequences because S2P cleaves not only BBF2H7 but also various transmembrane proteins, including ATF6 and the sterol regulatory element binding protein [54,55,56]. To establish the BBF2H7-Sec23a axis as a potential therapeutic target for pigment disorders, a comprehensive search for BBF2H7-specific activators or inhibitors is an important study that should be pursued in the future.

One limitation of this study is the insufficient in vivo investigation using *Bbf2h7* transgenic mice. In addition to the difficulty of detecting melanin pigmentation in neonatal mouse skin without stimuli such as UV irradiation, a critical obstacle is that *Bbf2h7*–/– mice only survive briefly after birth [25], which restricts in vivo analysis. Future studies should apply model systems such as melanocyte-specific *Bbf2h7* deletion mice or the inducible deletion of *Bbf2h7* in adult mice via a tamoxifen-inducible Cre/loxP system [57]. Validating our in vitro findings in vivo may also provide a more precise understanding of melanin pigmentation mediated by ER-derived signaling.

Another limitation of the present study is that we only investigated the causal relationship between BBF2H7 and Tyrosinase using B16 cells and primary melanocytes. Although we revealed that BBF2H7 does not directly affect the expression of Tyrosinase in these cells (Figure 2B and Figure 4B), analyzing expression levels using the patient-derived samples is essential. An important point to mention is that the gene expression comparison set we applied [41] does not indicate the regulation of Tyrosinase expression by BBF2H7, as shown in Figure 1B and Appendix A. Our present study suggests that the regulation of protein transport by BBF2H7, which is unrelated to the direct expression and enzymatic activity of Tyrosinase, is important for melanogenesis. Further research to examine BBF2H7 and Tyrosinase expression in melanocytes derived from patients with pigmentation disorders may elucidate the precise role of protein transport mediated by BBF2H7, as well as the expression level of Tyrosinase and ER folding capacity in melanin production. Moreover, the melanogenesis mechanism differs between melanoma and normal melanocytes. In this study, we clarified the importance of anterograde transport of Tyrosinase proteins, which are regulated by BBF2H7 for melanogenesis in the melanoma cell line and primary melanocytes extracted from neonatal mice as the first step. The same experiment for normal human skin melanocytes will be a critical step for precisely understanding the relationship between melanogenesis and BBF2H7.

In conclusion, our study demonstrates that COPII-mediated transport of Tyrosinase is essential for melanin production. Since these processes are regulated by ER-derived signals, protein transport machinery is also crucial and a fundamental ER function in the regulation of melanogenesis, in addition to protein quality control. A more advanced understanding of ER-derived signaling, including the UPR, may elucidate the overall molecular mechanisms governing melanogenesis and offer prospective therapeutic targets for melanin pigmentation disorders.

## 4. Materials and Methods

### 4.1. Mice

*Bbf2h7*^−/−^ mice on a C57BL/6 background were generated as previously described [25]. Littermate embryos at embryonic day 18.5 (sex not determined) were obtained from six pregnant mice over a five-month period at Kanazawa University. Mice were maintained under a 12 h light/dark cycle, and bedding (wood chips) was replaced weekly. For the collection of fetal mice, pregnant mice were humanely euthanized by cervical dislocation.

For the investigation of phenotypes of *Bbf2h7*^−/−^ mice, littermate wild-type embryos were used as controls. The following numbers of embryos were used for histological analyses, focusing on morphological alterations and melanin pigmentation: four wild-type, five *Bbf2h7*^−/−^, and thirteen *Bbf2h7^+^*^/−^ mice. For primary melanocyte cultures, eight wild-type, fourteen *Bbf2h7*^−/−^, and eighteen *Bbf2h7^+^*^/−^ embryos were used. Together with the six pregnant mice, a total of 68 mice were utilized. Because *Bbf2h7^+^*^/−^ mice were not included in the comparison groups, these mice were excluded from the analyses. To perform at least three independent experiments and to obtain sufficient cell numbers from embryonic samples with genotypes that could not be predetermined, a larger number of mice than initially planned was required. Throughout all procedures, A.S. was aware of group allocation, whereas the experimenter conducting the assays was blinded to genotype. All experiments were approved by the Institutional Animal Care and Use Committee of Kanazawa University (AP24-057, Approval Date: 15 October 2024).

### 4.2. Cell Culture and Treatments

B16 mouse melanoma cells (#RCB1283, JCRB Cell Bank, Ibaraki, Japan) were maintained in Dulbecco’s modified Eagle’s medium (Wako, Tokyo, Japan) supplemented with 10% fetal bovine serum (Biosera, Kansas City, MO, USA) at 37 °C in 5% CO_2_. Primary mouse melanocytes were extracted and cultured with modifications to previously described methods [58,59]. Briefly, cells were isolated from the dorsal skin of embryonic day 18.5 mice. The skin was excised from the dorsolateral region of the trunk between the limbs and incubated in 0.25% trypsin (Gibco, Waltham, MA, USA) for 3 h at 37 °C. Epidermal sheets were mechanically separated from the dermis using fine forceps and minced with a sterile razor blade. After adding a 10 mg/mL trypsin inhibitor (Sigma-Aldrich, St. Louis, MO, USA), the cell suspension was passed through a sterile 30 μm cell strainer (Sysmex, Kobe, Japan). Cell pellets were resuspended in Ham’s F-10 nutrient mix (Gibco) and supplemented with 10 μg/mL insulin (Sigma-Aldrich), 0.5 mg/mL bovine serum albumin (Sigma-Aldrich), 1 μM ethanolamine (Wako), 1 μM phosphoethanolamine (Sigma-Aldrich), 0.5 mM dibutyryl cAMP (Selleck, Tokyo, Japan), 70 μg/mL benzylpenicillin potassium (Wako), 100 μg/mL streptomycin sulfate (Wako), 50 μg/mL gentamycin sulfate (Wako), and 0.25 μg/mL amphotericin B (Sigma-Aldrich). Cells were maintained on 3.5 mg/mL collagen (Corning, Corning, NY, USA)-coated dishes for 16 days. To induce melanogenesis, cells were treated with 200 nM α-MSH (Sigma-Aldrich) for 24 or 48 h. Recombinant adenoviruses expressing mouse *Sec23a* or an empty vector were generated as previously described [25]. Adenoviral vectors expressing mouse BBF2H7 were constructed using the AdenoX Expression System (Takara USA, San Jose, CA, USA) according to the manufacturer’s instructions. Cells were infected with adenoviruses 30 h prior to analysis. To inhibit IRE1 activity, cells were treated with 15 μM 4μ8c (Sigma-Aldrich) for 54 h. For knockdown experiments, cells were transfected with siRNAs targeting mouse *Bbf2h7* (s101905, Thermo Fisher Scientific, Waltham, MA, USA), mouse *Sec23a* (s73500, Thermo Fisher Scientific), mouse *Atf6* (s105470, Thermo Fisher Scientific), or the non-targeting control siRNA (#4390843, Thermo Fisher Scientific) using ScreenFect siRNA reagent (Wako). Transfected cells were incubated for 30 h prior to α-MSH treatment.

### 4.3. Protein Preparation and Western Blotting

Proteins were extracted from B16 cells and primary melanocytes using the RIPA buffer (150 mM NaCl, 1.0% NP-40, 0.5% sodium deoxycholate, 0.1% SDS, 50 mM Tris-HCl) and supplemented with Protease Inhibitor Cocktail Set V (Wako) and Phosphatase Inhibitor Cocktail 3 (Sigma-Aldrich) at 4 °C. Lysates were centrifuged at 15,000× *g* for 15 min, and protein concentrations were determined using a bicinchoninic acid assay kit (Thermo Fisher Scientific). Equal amounts of protein (10 μg for Tyrosinase, 1 μg for β-actin, and 5 μg for others) were subjected to SDS–PAGE. The following primary antibodies were used for immunoblotting: anti-β-actin (1:100,000; Sigma-Aldrich), anti-Tyrosinase (1:1000; Abcam, Cambridge, UK), anti-Sec23a (1:1000; Sigma-Aldrich), anti-IRE1 (1:1000; Cell Signaling Technology, Danvers, MA, USA), anti-XBP1s (1:1000; Cell Signaling), and anti-ATF6 (1:1000; Novus Biologicals, Centennial, CO, USA). An anti-BBF2H7 antibody recognizing the N-terminal region (amino acids 1–292) (1:1000) was generated as previously described [39].

### 4.4. Immunofluorescence Staining

B16 cells and primary melanocytes were fixed with 4% paraformaldehyde for 1 h, followed by cold methanol for 30 min. Cells were permeabilized with 0.1% Triton X-100 for 5 min. The following primary antibodies were used: anti-BBF2H7 (1:500), anti-Tyrosinase (1:500), anti-Sec31a (1:500; BD Biosciences, Milpitas, CA, USA), and anti-KDEL (1:500; ENZO Life Sciences, Farmingdale, NY, USA). Nuclei were stained with 4′,6-diamidino-2-phenylindole (Thermo Fisher Scientific), and cells were imaged using confocal microscopy (FV1000D; Evident, Tokyo, Japan). Image processing and analysis of Tyrosinase staining were performed using ImageJ 1.54p (Java 1.8.0_172 64-bit) (accessed on 12 November 2025) (NIH, Rockville, MD, USA) [60]. Binary images were generated using the “Threshold” function, and Tyrosinase-positive foci were quantified using “Analyze Particles”.

### 4.5. Chromatin Immunoprecipitation Assay

Chromatin immunoprecipitation (ChIP) assays were performed using a ChIP Assay kit (Sigma-Aldrich) according to the manufacturer’s instructions. Briefly, B16 cells were cross-linked using 1% formaldehyde for 15 min at 37 °C. After inactivation with 0.15 M glycine for 5 min at room temperature, the cells were lysed with an SDS lysis buffer and sonicated (10 × 5 s of sonication pulses at 1 min intervals) (Sonifier 250; Emerson, Danbury, CT, USA). Equal amounts of chromatin from each sample were incubated overnight at 4 °C with anti-BBF2H7 antibody. Cross-linking was reversed by incubating for 6 h at 65 °C. DNA was purified by phenol–chloroform extraction and ethanol precipitation. Purified DNA was used for PCR analysis, and quantitative PCR was performed using the KAPA SYBR Fast qPCR kit (Roche, Basel, Switzerland) and a Light Cycler 480 system II (Roche). The primers used to detect the mouse *Sec23a* promoter were 5′-CTCATTAGGTAGCTCAAGGAGTCTC-3′ (forward) and 5′-CACTCGGCTAGTGGTGATGGTTCATG-3′ (reverse).

### 4.6. Melanin Content Assay

Cells were lysed in 2 M NaOH containing 10% dimethyl sulfoxide at 85 °C for 10 h. Melanin content was quantified by measuring absorbance at 405 nm using a MULTISKAN FC microplate reader (Thermo Fisher Scientific), and phase-contrast images were captured using confocal microscopy (FV1000D; Evident).

### 4.7. Histological Analysis

Abdominal skin from *Bbf2h7*^+/+^ and *Bbf2h7*^−/−^ mice on embryonic day 18.5 was fixed in 10% formalin. Hematoxylin–eosin staining was performed on 6 μm paraffin-embedded sections using standard protocols. Fontana–Masson staining was performed using a commercial kit (ScyTek Laboratories, Logan, UT, USA) according to the manufacturer’s instructions; stained tissues were imaged using a BX51 microscope (Evident).

### 4.8. Data Collection

Gene expression data were obtained from human-embryonic-stem-cell-derived melanocytes and melanocytes in patients with oculocutaneous albinism from the Gene Expression Omnibus (NCBI, Bethesda, MD, USA) (GSE45226) [41]. Genes exhibiting more than a two-fold increase or decrease in expression (*p* < 0.05) between the two samples were selected for analysis. Pathway and gene ontology analyses were performed using ShinyGo 0.85 (http://bioinformatics.sdstate.edu/go/) (accessed on 10 November 2025) [61].

### 4.9. Statistical Analysis

Statistical comparisons were performed using one-way ANOVA, followed by Tukey’s post hoc test. Box plots were generated using the BoxPlotR web tool (http://shiny.chemgrid.org/boxplotr/) (accessed on 12 November 2025) [62]. Differences were considered statistically significant at *p* < 0.05, and *p*-values less than 0.05 and 0.01 are indicated as * and **, respectively.

## Figures and Tables

**Figure 1 ijms-27-00501-f001:**
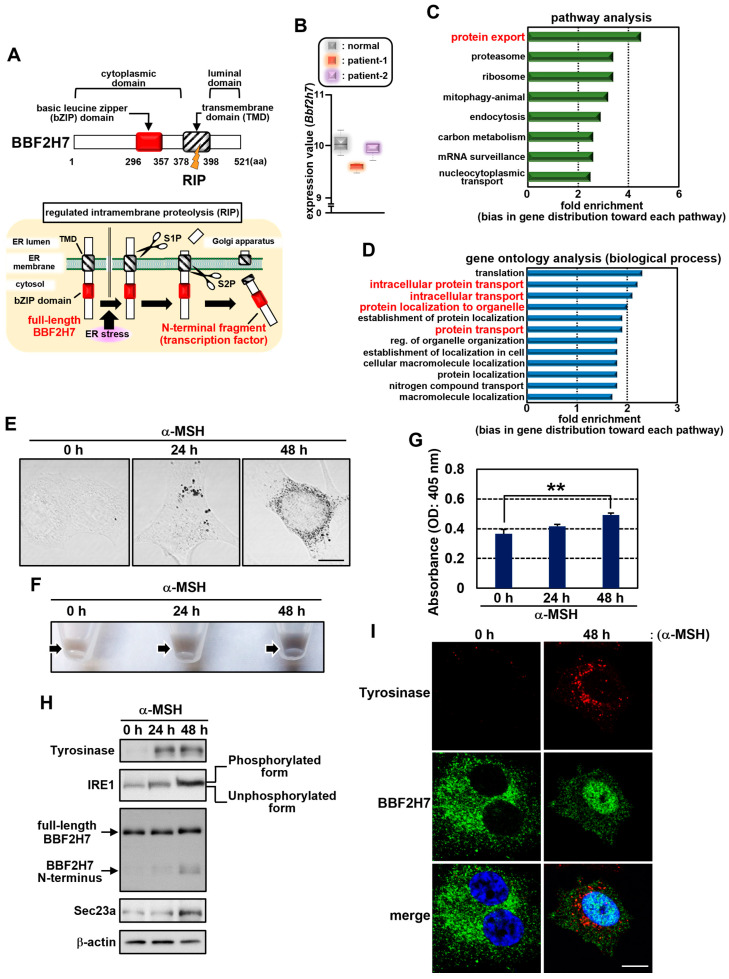
The ER stress sensor BBF2H7 is activated during melanogenesis. (**A**) Schematic representation of the putative structure (upper) and activation mechanism (lower) of BBF2H7. BBF2H7 is an ER-resident transmembrane transcription factor that translocates to the Golgi apparatus in response to ER stress, where it is cleaved by regulated intramembrane proteolysis (RIP) via site-1 and site-2 proteases (S1P and S2P). The generated cytoplasmic N-terminal fragment, containing a basic leucine zipper (bZIP) DNA-binding domain, functions as a transcription factor. (**B**) Expression levels of *Bbf2h7* in melanocytes derived from human embryonic stem cells (normal) and in melanocytes derived from pluripotent stem cells of patients with oculocutaneous albinism (patient-1 and patient-2); data were obtained from the Gene Expression Omnibus Dataset GSE45226 and are shown as a box plot (*n* = 3). (**C**) Pathway analysis of genes showing more than a two-fold increase or decrease in expression in patient-1 relative to normal melanocytes (*p* < 0.05) in GSE45226. “Protein export” is the most enriched category (fold enrichment: 4.5). (**D**) Gene ontology (biological process) analysis of differentially expressed genes (as in (**C**)). The second-, third-, fourth-, and sixth-largest categories are “intracellular protein transport” (2.2-fold), “intracellular transport” (2.1-fold), “protein localization to organelle” (2.0-fold), and “protein transport” (1.9-fold), respectively. (**E**) Phase-contrast images of B16 cells treated with α-MSH. Scale bar: 10 μm. (**F**) Cell pellets (arrows) of B16 cells treated with α-MSH. (**G**) Melanin content in B16 cells treated with α-MSH (*n* = 3). (**H**) Western blot analysis of Tyrosinase, IRE1, BBF2H7, and Sec23a in B16 cells treated with α-MSH. (**I**) Immunofluorescence staining of Tyrosinase (red) and BBF2H7 (green) in B16 cells treated with α-MSH (DAPI: blue). Scale bar: 10 μm. Mean ± SD. ** *p* < 0.01.

**Figure 2 ijms-27-00501-f002:**
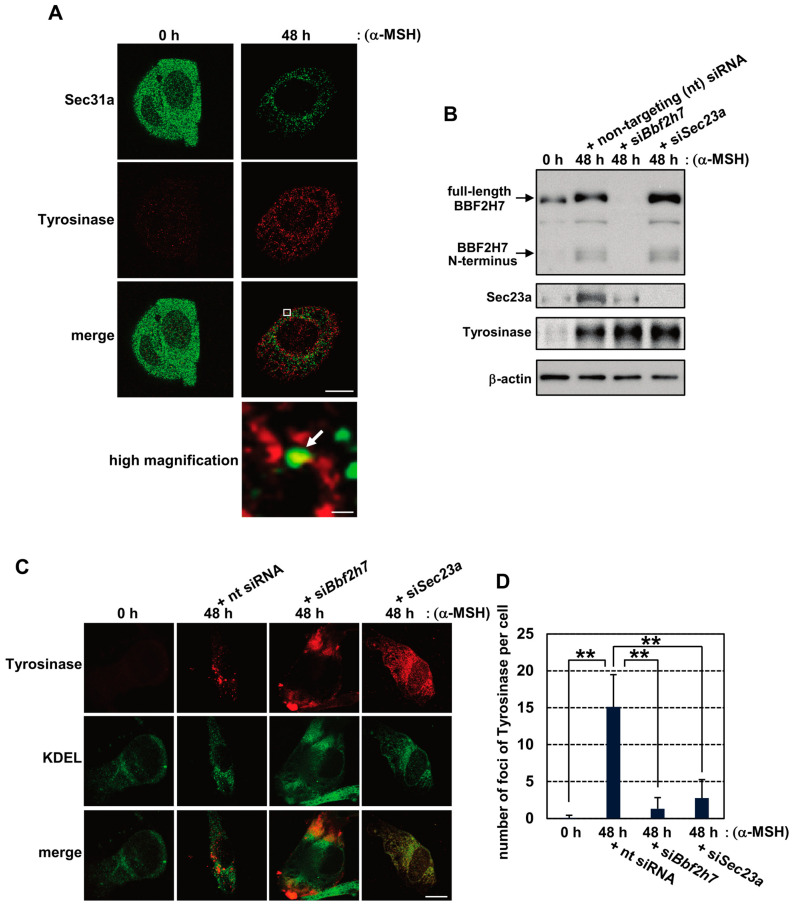
BBF2H7 accelerates vesicular transport of Tyrosinase during melanogenesis. (**A**) Immunofluorescence staining of Sec31a (green) and Tyrosinase (red) in B16 cells treated with α-MSH. The region in the white box is shown at higher magnification. Arrow indicates the colocalized Sec31a and Tyrosinase signals. Scale bars: 10 μm (merged image) and 300 nm (high magnification). (**B**) Western blot analysis of BBF2H7, Sec23a, and Tyrosinase in B16 cells treated with α-MSH and transfected with non-targeting (nt) siRNA, si*Bbf2h7*, or si*Sec23a*. (**C**) Immunofluorescence staining of Tyrosinase (red) and KDEL (ER marker) (green) in B16 cells treated with α-MSH and transfected with nt siRNA, si*Bbf2h7*, or si*Sec23a*. Scale bar: 10 μm. (**D**) Quantification of Tyrosinase-positive foci in (**C**), measured using ImageJ 1.54p (Java 1.8.0_172 64-bit) (*n* = 9). Mean ± SD. ** *p* < 0.01.

**Figure 3 ijms-27-00501-f003:**
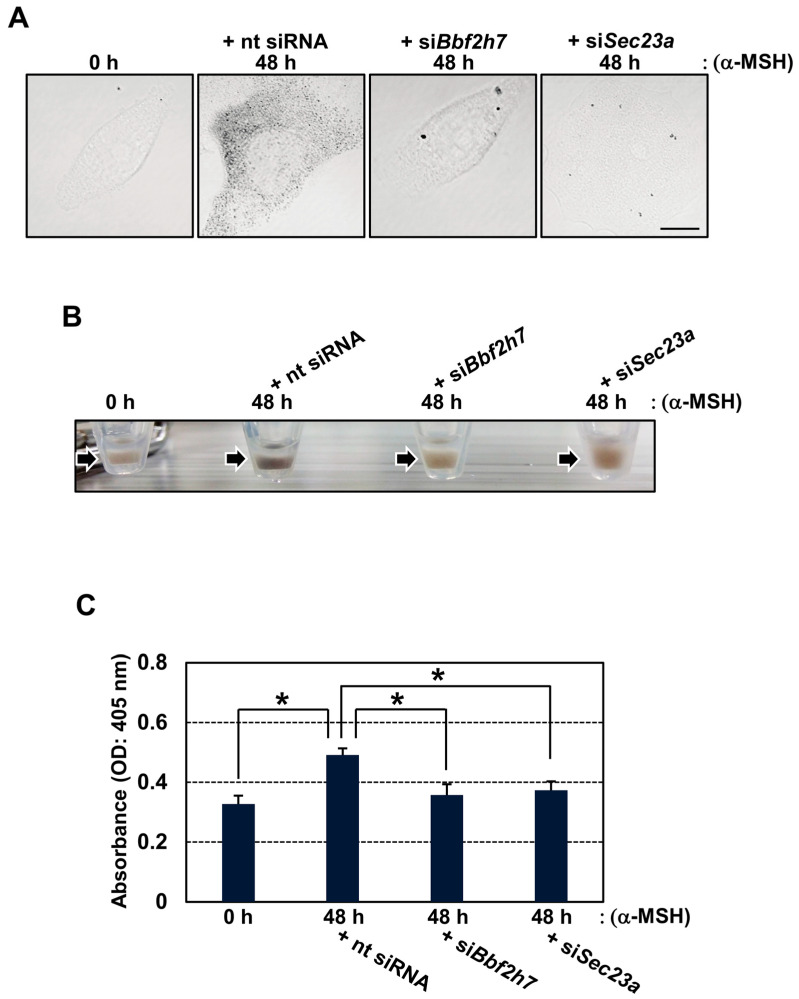
BBF2H7-dependent vesicular transport is required for melanin production. (**A**) Phase-contrast images of B16 cells treated with α-MSH and transfected with nt siRNA, si*Bbf2h7*, or si*Sec23a*. Scale bar: 10 μm. (**B**) Cell pellets (arrows) of B16 cells treated with α-MSH and transfected with nt siRNA, si*Bbf2h7*, or si*Sec23a*. (**C**) Melanin content in B16 cells treated with α-MSH and transfected with nt siRNA, si*Bbf2h7*, or si*Sec23a* (*n* = 3). Mean ± SD. * *p* < 0.05.

**Figure 4 ijms-27-00501-f004:**
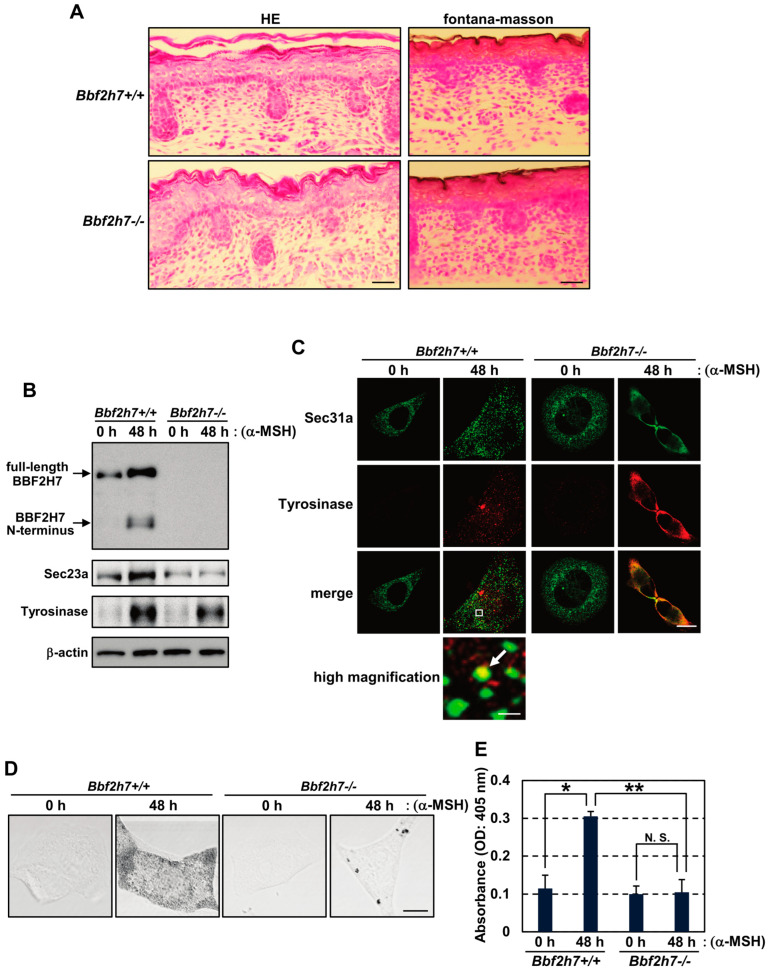
Vesicular transport of Tyrosinase and melanin production are inhibited by *Bbf2h7* deficiency. (**A**) HE staining (left) and Fontana–Masson staining (right) of the abdominal epidermis and dermis of embryonic day 18.5 *Bbf2h7*^+/+^ and *Bbf2h7*^−/−^ mice. Scale bars: 50 μm. (**B**) Western blot analysis of BBF2H7, Sec23a, and Tyrosinase in primary melanocytes treated with α-MSH. (**C**) Immunofluorescence staining of Sec31a (green) and Tyrosinase (red) in primary melanocytes treated with α-MSH. The region in the white box is shown at higher magnification. Arrow indicates colocalized Sec31a and Tyrosinase signals. Scale bars: 10 μm (merged image) and 300 nm (high magnification). (**D**) Phase-contrast images of primary melanocytes treated with α-MSH. Scale bar: 10 μm. (**E**) Melanin content in primary melanocytes treated with α-MSH (*n* = 3). N.S., not significant. Mean ± SD. * *p* < 0.05; ** *p* < 0.01.

**Figure 5 ijms-27-00501-f005:**
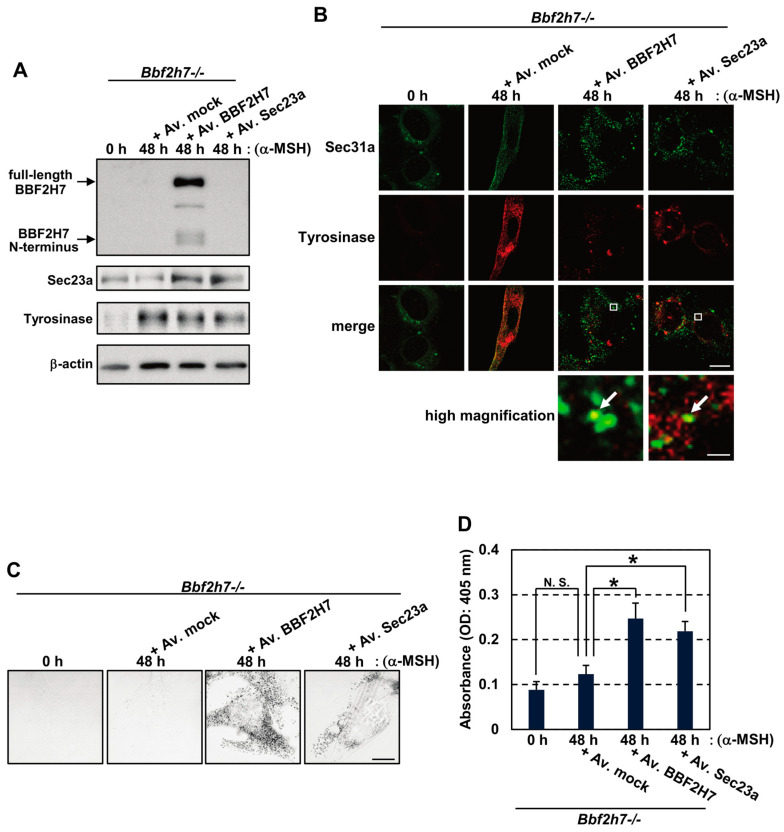
Expression of BBF2H7 or Sec23a rescues melanin production in *Bbf2h7^−/−^* melanocytes. (**A**) Western blot analysis of BBF2H7, Sec23a, and Tyrosinase in primary melanocytes treated with α-MSH and infected with adenovirus expressing BBF2H7, Sec23a, or an empty vector (mock). Av., adenovirus. (**B**) Immunofluorescence staining of Sec31a (green) and Tyrosinase (red) in primary melanocytes treated with α-MSH and infected with adenovirus-expressing BBF2H7 or Sec23a. The region in the white box is shown at higher magnification. Arrows indicate colocalized Sec31a and Tyrosinase signals. Scale bars: 10 μm (merged image) and 300 nm (high magnification). (**C**) Phase-contrast images of primary melanocytes treated with α-MSH and infected with adenovirus-expressing BBF2H7 or Sec23a. Scale bar: 10 μm. (**D**) Melanin content in primary melanocytes treated with α-MSH and infected with adenovirus-expressing BBF2H7 or Sec23a (*n* = 3). N.S., not significant. Mean ± SD. * *p* < 0.05.

## Data Availability

The original contributions presented in this study are included in the article/Appendix A. Further inquiries can be directed to the corresponding author.

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
