# Peer review of "Vesicular Transport Mediated by Endoplasmic Reticulum Stress Sensor BBF2H7 Orchestrates Melanin Production During Melanogenesis"

_ijms, 2026, doi:10.3390/ijms27010501_

Round 1

Reviewer 1 Report

Comments and Suggestions for Authors

In this study, the authors demonstrate that Box B-binding factor 2 human homolog on chromosome 7 (BBF2H7), an ER-resident transmembrane transcription factor that functions as an ER stress sensor, is activated by the mild ER stress caused by abundant Tyrosinase synthesis. Activated BBF2H7 enhances COPII-mediated anterograde transport by inducing the expression of Sec23a, a COPII component and transcriptional target of BBF2H7. However, there are some questions below that require modification and supplementation.

  1. The study only utilized public datasets and literature citations to indicate lower expression of Bbf2h7 in samples from patients with albinism and vitiligo. However, it did not directly assess a causal relationship between the BBF2H7 and Tyrosinase expression in melanocytes from these patients. This inference can be further analyzed and discussed in conjunction with the authors' research data on "BBF2H7-tyrosinase relationship" presented below.
  2. It did not analyze the synergistic or antagonistic relationship between BBF2H7 and other ER stress sensors such as IRE1 and ATF6.
  3. The B16 cell line is derived from melanoma, and its melanogenesis mechanism differs from that of normal melanocytes. Moreover, primary melanocytes from embryonic mice have not undergone physiological maturation and may not fully replicate pigment regulation in adult skin. The study did not cross-validate its findings using multiple models, such as normal human melanocytes.
  4. Although the article proves the BBF2H7-Sec23a axis as a potential therapeutic target for pigment disorders, it did not conduct relevant experiments involving the application of BBF2H7 activators or inhibitors.

Author Response

Comments 1: The study only utilized public datasets and literature citations to indicate lower expression of Bbf2h7 in samples from patients with albinism and vitiligo. However, it did not directly assess a causal relationship between the BBF2H7 and Tyrosinase expression in melanocytes from these patients. This inference can be further analyzed and discussed in conjunction with the authors' research data on "BBF2H7-tyrosinase relationship" presented below.

Response 1: Thank you for pointing this out. We agree with this comment. A limitation of the present study is that we only investigated the causal relationship between BBF2H7 and Tyrosinase using B16 cells and primary melanocytes. Although we found that BBF2H7 does not directly affect the expression of Tyrosinase in these cells (Figures 2B and 4B), the analysis of the expression levels using the patient-derived samples is essential. Importantly, the gene expression comparison set we applied does not indicate the regulation of Tyrosinase expression by BBF2H7. Indeed, the level of Tyrosinase expression was not reduced in patients with oculocutaneous albinism, uncorrelated with that of BBF2H7 (Supplemental Figure 1). Our present study suggests that the regulation of protein transport by BBF2H7, which is unrelated to the direct expression and enzymatic activity of Tyrosinase, is important for melanogenesis. Further research to examine BBF2H7 and Tyrosinase expression in melanocytes derived from the patients with pigmentation disorders may elucidate the precise contribution of protein transport mediated by BBF2H7, the expression level of Tyrosinase and the endoplasmic reticulum (ER)’s folding capacity to melanin production. We have added these descriptions to the revised manuscript (page 3, lines 123–124; page 13, lines 336–347; and Supplemental Figure 1).

Comments 2: It did not analyze the synergistic or antagonistic relationship between BBF2H7 and other ER stress sensors such as IRE1 and ATF6.

Response 2: We confirmed that the expression levels of IRE1 and ATF6 were not significantly changed by the knockdown of Bbf2h7 (Supplemental Figure 4A), suggesting that there is no crosstalk between the BBF2H7 pathway and the IRE1 or ATF6 pathway. Additionally, we carried out melanin content assays using Bbf2h7-knockdown B16 cells treated with 4m8c (a specific inhibitor of IRE1) or siRNA targeting Atf6. Melanin production was not synergistically reduced by the suppression of both the BBF2H7 pathway and the IRE1 or ATF6 pathway (Supplemental Figures 4B, 4C, 4D). We previously revealed that the downregulation of ER folding capacity by the inhibition of IRE1 or ATF6 pathway suppressed melanin production (Yamazaki et al., Journal of Dermatological Science, 2025, doi: 10.1016/j.jdermsci.2025.01.001). Taken together, the inhibition of either anterograde transport of Tyrosinase (BBF2H7 pathway) or protein quality control in the ER lumen (IRE1 and ATF6 pathways) is sufficient to attenuate melanin production. We have added these findings to the revised manuscript (page 7, lines 210–223, and Supplemental Figure 4).

Comments 3: The B16 cell line is derived from melanoma, and its melanogenesis mechanism differs from that of normal melanocytes. Moreover, primary melanocytes from embryonic mice have not undergone physiological maturation and may not fully replicate pigment regulation in adult skin. The study did not cross-validate its findings using multiple models, such as normal human melanocytes.

Response 3: We appreciate your comment. As you mentioned above, the melanogenesis mechanism differs between melanoma and normal melanocytes. The same experiment toward normal human skin melanocytes is a crucially important step for precisely understanding the relationship between melanogenesis and BBF2H7. Moreover, primary melanocytes from embryonic mice may not fully replicate pigment regulation in adult skin, and external stimuli such as UV exposure are generally required to assess melanin pigments. Future studies applying model systems such as melanocyte-specific Bbf2h7 deletion mice or the inducible deletion of Bbf2h7 in adult mice via a tamoxifen-inducible Cre/loxP system will be necessary. In this study, we clarified the importance of anterograde transport of Tyrosinase protein regulated by BBF2H7 for melanogenesis in the melanoma cell line and primary melanocytes extracted from neonatal mice as a first step. Verifying similar results in human cells will be an essential step toward developing a therapeutic approach. We have discussed this point as a next step which should be resolved in the future (page 13, lines 328–335 and lines 347–353).

Comments 4: Although the article proves the BBF2H7-Sec23a axis as a potential therapeutic target for pigment disorders, it did not conduct relevant experiments involving the application of BBF2H7 activators or inhibitors.

Response 4: We appreciate this pertinent suggestion. We understand the importance of the application of specific activators or inhibitors for BBF2H7. However, unfortunately, such compounds have yet to be discovered. BBF2H7 is activated as a transcription factor via cleavage by site-1 and site-2 proteases (Figure 1A). Several compounds have been shown to potentially alter the activity of BBF2H7. For example, a previous report has indicated that Nelfinavir inhibits site-2 protease activity (Guan et al., FEBS Journal, 2012, doi: 10.1111/j.1742-4658.2012.08619.x). However, treatment with Nelfinavir may have unintended consequences, because the site-2 protease cleaves not only BBF2H7 but also various transmembrane proteins including ATF6 and sterol regulatory element-binding protein (Duncan et al., Journal of Biological Chemistry, 1997, doi: 10.1074/jbc.272.19.12778; Rawson et al., Molecular Cell, 1997, doi: 10.1016/s1097-2765(00)80006-4; Ye et al., Molecular Cell, 2000, doi: 10.1016/s1097-2765(00)00133-7). To eliminate such unexpected effects, we have been forced to use siRNA targeting Bbf2h7 in this study. To establish the BBF2H7-Sec23a axis as a potential therapeutic target for pigment disorders, a comprehensive search for BBF2H7-specific activators or inhibitors is an extremely important study that should be pursued in the future. We have discussed this point in the revised manuscript (pages 12–13, lines 320–327).

Response to Comments on the Quality of English Language

Point 1: The English could be improved to more clearly express the research.

Response 1: We apologize for this confusion. We have improved the English by using the English Editing Services of MDPI Author Services.

Additional clarifications

We have added new experimental procedures in section “Materials and Methods” (page 14, line 405, line 408, and lines 421–422 and page 15, lines 435–448).

Reviewer 2 Report

Comments and Suggestions for Authors

The manuscript reports a straightforward experimental approach showing that the ER Stress Sensor BBF2H7 is involved in tyrosinase transport the ER to the melanosomes, as it is a transcription factor upregulating Sec23a, a COPII component involved in anterograde transport. The manuscript is quite clear and findings interesting, but it would benefit of more information on the discovery and the role of Bbf2h7. Other points also need to be addressed (see below).

Major points

In the Introduction, it is not fully specified if BBF2H7 targets, as transcription factors, proteins other than tyrosinase

Line 78 The definition of “…a type II transmembrane protein localized 78 to the ER membrane that functions as an ER stress sensor” should be integrated by a few details on structure and orientation of the protein

Figure 1 Dimensions are not insufficient, and the panels are too close

Line 101 Authors should specify if the proteases involved in transmembrane proteolysis “intramembrane proteolysis mediated by site-1 and site-2 proteases” are known or unknown

Line 105 The sentence “…the gene expression comparison shown by Dr. Yvonne Mica and colleagues exhibited relatively low expression of 106 Bbf2h7 in melanocytes” is difficult to catch and the reference is unclear

Line 148 The conclusion that “BBF2H7 acts as a transcription factor and promotes Sec23a expression in response to ER stress” is not justified by results shown in Figure 1, as immunoblotting show that the same stimulus induces both the processing of BBF2H7 and higher expression of Sec23a, but does not show a direct binding of the transcription factor on Sec23a promoter. The sentence should be modified

Line 159 The conclusion that “Immunoreactivity of Tyrosinase partially overlapped with Sec31a-positive puncta in α-MSH-treated cells, suggesting that Tyrosinase is exported from the ER via COPII-dependent trafficking” should be supported by a sentence explaining the reason underlying only such a partial overlapping

The sentence “The amount of Sec23a was increased in α-MSH-treated Bbf2h7–/–  melanocytes exogenously expressing BBF2H7 or Sec23a” is unclear, because if Sec23a is exogenously expressed, its level is increased, independently from α-MSH

Minor points

Line 55 Please check the English of “…followed by inducing the expression…”

Line 62 Please check the English of the sentence  “These series of steps indicate that the ER function to enable 62 large-scale protein synthesis and its efficient transport from the ER to melanosomes is essential for melanin production, in addition to enzymatic activity of Tyrosinase”

Line 138 Please check the English of the sentence “…we applied B16 melanoma cell line treated with α-MSH”

Author Response

Comments 1: In the Introduction, it is not fully specified if BBF2H7 targets, as transcription factors, proteins other than tyrosinase.

Response 1: We apologize for the insufficient explanation. One of the targets of BBF2H7 is Sec23a, a key component of COPII transport vesicles responsible for anterograde trafficking from the endoplasmic reticulum (ER) to the Golgi apparatus. Activated BBF2H7 as a transcription factor promotes protein export from the ER via the direct induction of Sec23a (Saito et al., Nature Cell Biology, 2009, doi: 10.1038/ncb1962). Another target gene of BBF2H7 is activating transcription factor 5 (Atf5) (Izumi et al., Journal of Biological Chemistry, 2012, doi: 10.1074/jbc.M112.373746). The upregulation of ATF5 by BBF2H7 suppresses apoptosis. Of those, the downregulation of BBF2H7 leads to severe chondrodysplasia due to the accumulation of cartilage matrix proteins in the ER, suggesting that the regulation of protein transport by the BBF2H7-Sec23a pathway is important for maintaining biological functions. We have added these descriptions to the revised manuscript (pages 2–3, lines 87–94).

Comments 2: Line 78 The definition of “…a type II transmembrane protein localized 78 to the ER membrane that functions as an ER stress sensor” should be integrated by a few details on structure and orientation of the protein.

Response 2: Thank you for pointing this out. We agree with this comment. Under normal conditions, BBF2H7 is localized in the ER membrane. The C-terminal domain targets the ER lumen, and the N-terminal domain targets the cytoplasm, which contains a basic leucine zipper DNA-binding domain (Kondo et al., Molecular and Cellular Biology, 2007, doi: 10.1128/MCB.01552-06). Although the detailed mechanisms are still unknown, BBF2H7 senses ER stress and activates as a transcription factor via regulated intramembrane proteolysis. We have added these descriptions to the revised manuscript (page 2, lines 78–83).

Comments 3: Figure 1 Dimensions are not insufficient, and the panels are too close.

Response 3: We apologize for this confusion. We improved the dimensions and panel placement of Figure 1 in the revised manuscript.

Comments 4: Line 101 Authors should specify if the proteases involved in transmembrane proteolysis “intramembrane proteolysis mediated by site-1 and site-2 proteases” are known or unknown.

Response 4: Thank you for pointing this out. We previously demonstrated that BBF2H7 is cleaved by site-1 and site-2 proteases in response to ER stress (Kondo et al., Molecular and Cellular Biology, 2007, doi: 10.1128/MCB.01552-06). We have modified these descriptions, and indicated the reference paper in the revised manuscript (page 3, lines 107–109).

Comments 5: Line 105 The sentence “…the gene expression comparison shown by Dr. Yvonne Mica and colleagues exhibited relatively low expression of 106 Bbf2h7 in melanocytes” is difficult to catch and the reference is unclear.

Response 5: We sincerely apologize for this confusion. Dr. Yvonne Mica and colleagues obtained RNAs from melanocytes derived from human embryonic stem cells, and melanocytes derived from pluripotent stem cells of patients with oculocutaneous albinism. They performed genome-wide analysis of gene expression to compare the expression levels of various genes in normal and patient-derived melanocytes. Based on this data, we found that the expression level of Bbf2h7 was relatively low in patient-derived melanocytes, compared with those derived from human embryonic stem cells. We have added these descriptions to the revised manuscript (page 3, lines 115–123).

Comments 6: Line 148 The conclusion that “BBF2H7 acts as a transcription factor and promotes Sec23a expression in response to ER stress” is not justified by results shown in Figure 1, as immunoblotting show that the same stimulus induces both the processing of BBF2H7 and higher expression of Sec23a, but does not show a direct binding of the transcription factor on Sec23a promoter. The sentence should be modified.

Response 6: We previously revealed that BBF2H7 directly binds to the CRE-binding site within the Sec23a promoter, resulting in the induction of Sec23a expression in chondrocytes (Saito et al., Nature Cell Biology, 2009, doi: 10.1038/ncb1962). We also confirmed the direct binding of BBF2H7 to the Sec23a promoter in the B16 melanoma cell line treated with α-MSH by performing a chromatin immunoprecipitation assay (Supplemental Figure 3). We have added these findings to the revised manuscript (page 5, lines 161–165 and Supplemental Figure 3).

Comments 7: Line 159 The conclusion that “Immunoreactivity of Tyrosinase partially overlapped with Sec31a-positive puncta in α-MSH-treated cells, suggesting that Tyrosinase is exported from the ER via COPII-dependent trafficking” should be supported by a sentence explaining the reason underlying only such a partial overlapping.

Response 7: Thank you for pointing this out. We understand the importance of this comment. Tyrosinase proteins rapidly move from the ER into melanosomes. Thus, the majority of Tyrosinase-positive puncta in the cytoplasm are considered to be melanosomes. We may detect partial Sec31a- and Tyrosinase-positive puncta as COPII vesicles, containing the Tyrosinase protein. We have added these descriptions to the revised manuscript (page 5, lines 179–183).

Comments 8: The sentence “The amount of Sec23a was increased in α-MSH-treated Bbf2h7–/–  melanocytes exogenously expressing BBF2H7 or Sec23a” is unclear, because if Sec23a is exogenously expressed, its level is increased, independently from α-MSH.

Response 8: We apologize for the unclear explanation. We have modified the sentence, “The amount of Sec23a was increased in α-MSH-treated Bbf2h7–/– melanocytes exogenously expressing BBF2H7 or Sec23a” to “We observed an increase in the amount of Sec23a due to its exogenous expression. The induction of Sec23a was also observed in Bbf2h7–/– melanocytes; they exogenously expressed BBF2H7” (page 11, lines 267–269).

Comments 9: Line 55 Please check the English of “…followed by inducing the expression….

Response 9: Thank you for pointing this out. We rewrote the sentence, “Upon UV exposure, keratinocytes release α-MSH, which binds to the melanocortin 1 receptor on the plasma membrane of melanocytes, followed by inducing the expression of the transcription factor microphthalmia-associated transcription factor (MITF)” to “Upon UV exposure, keratinocytes release α-MSH, which binds to the melanocortin 1 receptor on the plasma membrane of melanocytes. This binding induces the expression of the microphthalmia-associated transcription factor (MITF)” (page 2, lines 53–56).

Comments 10: Line 62 Please check the English of the sentence “These series of steps indicate that the ER function to enable 62 large-scale protein synthesis and its efficient transport from the ER to melanosomes is essential for melanin production, in addition to enzymatic activity of Tyrosinase”.

Response 10: We rewrote the sentence, “These series of steps indicate that the ER function to enable large-scale protein synthesis and its efficient transport from the ER to melanosomes is essential for melanin production, in addition to enzymatic activity of Tyrosinase” to “This series of steps indicates that the ER function ensures large-scale protein synthesis, and the efficient transport of Tyrosinase protein from the ER to melanosomes is essential for melanin production” (page 2, lines 62–64).

Comments 11: Line 138 Please check the English of the sentence “…we applied B16 melanoma cell line treated with α-MSH”

Response 11: We rewrote the sentence, “Since BBF2H7 is activated in response to ER stress, we applied B16 melanoma cell line treated with α-MSH; a model in which we previously demonstrated that ER stress due to the abundant synthesis of Tyrosinase occurs during melanogenesis” to “ER stress is caused by the over-synthesis of Tyrosinase, and we previously demonstrated that this occurs due to the treatment of the B16 melanoma cell line with α-MSH. Since BBF2H7 is activated in response to ER stress, we used this model to investigate the relationship between BBF2H7 and melanogenesis.” (page 5, lines 152–155).

Response to Comments on the Quality of English Language

Point 1: We improved the English by using the English Editing Services of MDPI Author Services.

Additional clarifications

We have added new experimental procedures in the “Materials and Methods” section (page 14, line 405, line 408, and lines 421–422 and page 15, lines 435–448).

Round 2

Reviewer 1 Report

Comments and Suggestions for Authors

Accept 

Reviewer 2 Report

Comments and Suggestions for Authors

All issues raised have been satisfactorily addressed